# Nutrient Therapy for the Improvement of Fatigue Symptoms

**DOI:** 10.3390/nu15092154

**Published:** 2023-04-30

**Authors:** Michael Barnish, Mahsa Sheikh, Andrew Scholey

**Affiliations:** 1REVIV Life Science Research, REVIV Global Ltd., Manchester M15 4PS, UK; mbarnish@revivme.com; 2Centre for Human Psychopharmacology, Swinburne University, Melbourne, VIC 3122, Australia; andrew@scholeylab.com; 3Department of Nutrition, Dietetics and Food, Monash University, Notting Hill, VIC 3168, Australia

**Keywords:** vitamin supplementation, fatigue, fatigue symptoms, nutrient therapy, nootropic, supplementation

## Abstract

Fatigue, characterised by lack of energy, mental exhaustion and poor muscle endurance which do not recover following a period of rest, is a common characteristic symptom of several conditions and negatively impacts the quality of life of those affected. Fatigue is often a symptom of concern for people suffering from conditions such as fibromyalgia, chronic fatigue syndrome, cancer, and multiple sclerosis. Vitamins and minerals, playing essential roles in a variety of basic metabolic pathways that support fundamental cellular functions, may be important in mitigating physical and mental fatigue. Several studies have examined the potential benefits of nutrients on fatigue in various populations. The current review aimed to gather the existing literature exploring different nutrients’ effects on fatigue. From the searches of the literature conducted in PubMed, Ovid, Web of Science, and Google scholar, 60 articles met the inclusion criteria and were included in the review. Among the included studies, 50 showed significant beneficial effects (*p* < 0.05) of vitamin and mineral supplementation on fatigue. Altogether, the included studies investigated oral or parenteral administration of nutrients including Coenzyme Q10, L-carnitine, zinc, methionine, nicotinamide adenine dinucleotide (NAD), and vitamins C, D and B. In conclusion, the results of the literature review suggest that these nutrients have potentially significant benefits in reducing fatigue in healthy individuals as well as those with chronic illness, both when taken orally and parenterally. Further studies should explore these novel therapies, both as adjunctive treatments and as sole interventions.

## 1. Introduction

Fatigue is amongst common complaints in daily life, negatively impacting work performance, family life, and social relationships. Fatigue can be conceptualised as a sensation of exhaustion or difficulty performing intellectual or physical activities, which is often not recovered after a period of rest [1]. Patients with fatigue typically report lack of energy, mental exhaustion, poor muscle endurance, delayed recovery after physical exertion, and nonrestorative sleep [2]. Mood scales which include fatigue measures such as Bond-Lader Visual Analogue Scale (BL-VAS) and Profile of Mood Scale (POMS) are typically used for measuring fatigue states. Moreover, several fatigue scales including the Fatigue Assessment Scale (FAS), the Fatigue Severity Scale (FSS), and the Chalder Fatigue Scale (CFQ 11) are often used for measuring the nature and severity of fatigue.

While no pathology can be identified in one third of fatigue cases [2], several conditions including depression, viral illness, upper respiratory tract infection, anaemia, cancer, and lung disease, are amongst the common causes of fatigue. Therefore, fatigue can be classified as secondary to other medical conditions, which might last for a month or longer, in cases of physiologic and chronic fatigue, respectively [2]. Physiologic fatigue, characterized by imbalance in the routines of exercise, sleep, diet, or other activities unrelated to an underlying medical condition, is most common in adolescents and elderly and is relieved with rest [3]. However, chronic fatigue, characterised by intense physical and mental fatigue of unknown cause, is not relieved with rest and lasts longer than six months [2]. Accordingly, Myalgic Encephalomyelitis/Chronic Fatigue Syndrome (ME/CFS) is a disabling clinical condition, characterised by persistent post-exertional fatigue which produces various degrees of disability by limiting the patient’s functional capacity and is accompanied by a variety of symptoms associated with cognitive, immunological, endocrinological, and autonomous dysfunction [4]. Chronic Fatigue Syndrome (CFS), characterized by prolonged and relapsing fatigue, is a serious and extremely debilitating chronic illness that results in significant disability, lasting from 6 months to several years [5]. Moreover, CFS is associated with a wide range of physical and behavioural symptoms, including joint and muscle pain, headache, intolerance to physical exertion, anxiety, depression cognitive impairment and sleep disorders [3]. Several common medical conditions are known to cause fatigue symptoms. Fatigue is amongst the major side effects of cancer and its treatment, disrupting the quality of life of cancer patients and may be a risk factor in reduced survival [6]. Fatigue is also described by patients with multiple sclerosis as their most disturbing symptom [7]. Furthermore, fibromyalgia, a common chronic pain syndrome, associated with widespread pain at multiple tender points, is characterised by fatigue and sleep disruption [8]. Severe fatigue is reported by all fibromyalgia patients, with as many as 80% of patients with this condition also fulfilling the criteria for CFS [9].

Management of fatigue is complex and requires a multimodal therapeutic approach. Gradual physical exercise together with cognitive behaviour therapy, remain the most common approaches [2,3]. Vitamins and minerals, playing essential roles in a variety of basic metabolic pathways that support fundamental cellular functions, affect cognitive and psychological processes including mental and physical fatigue [10]. Therefore, several micronutrients, enzymes and amino acids have been suggested to have a role on fatigue, cognition, or psychological functions [10]. Efficient and safe vitamin and mineral supplementation of individuals suffering from fatigue, may provide a beneficial treatment option. The current review hence gathers studies investigating vitamin C, vitamin D, B vitamins, Coenzyme Q10 (CoQ10), L-carnitine, zinc, nicotinamide adenine dinucleotide (NADH), and methionine, all of which have been suggested as having impacts on physical and mental fatigue, in the management of fatigue symptoms in different conditions.

## 2. Methods

Searches of the literature were conducted in PubMed, Ovid (including Journals from Ovid, AMED (Allied and Complementary Medicine), Embase, Global health, and Ovid MEDLINE), Web of Science, and Google scholar. The search was conducted without any geographical or time restrictions using the following keywords: “vitamin”, “vitamin supplementation”, “IV therapy”, “intravenous therapy”, “IM therapy”, “Intramuscular therapy”, “oral supplementation”, “supplementation”, “nutrient therapy”, “vitamin therapy”, “micronutrient therapy”, “fatigue”, and “fatigue symptoms”. In order to achieve comprehensive review of the literature, these keywords were used in combination with the following vitamins and minerals: vitamins B, C, and D; glutathione, N-acetyl cysteine, alpha lipoic acid, manganese, copper, selenium, chromium, magnesium, zinc, methionine, inositol, leucine, L-carnitine, choline, L-taurine, CoQ10, and NAD. Studies employing both single nutrients as well as combinations of the mentioned micronutrients were included in the literature review. In addition, the most common intravenous nutritional intervention, the Myer’s cocktail, has also been included. The included micronutrients in this literature search are amongst the most widely used nutrients in the management of health conditions [11] as well as fatigue symptoms [10] and are also available for both oral and parenteral administration [12]. A manual search of all included bibliographies was carried out to identify any omitted articles. From the combinations of the keywords and relevant articles found in review references, 651 articles were identified. Studies that satisfied the inclusion criteria were retrieved for full-text assessment. Articles were considered for inclusion if they reported data from an original study examining the effects of a nutrient supplementation on fatigue status, performed statistical analysis, were accessible in their full text and published in English language. The exclusion criteria were as follows: conference proceedings, case reports, letters, summaries, expert opinions and comments, and articles published in a language other than English. Therefore, following full-text analysis of the 124 retrieved articles, 60 studies met the inclusion criteria and were included for analysis (Figure 1).

## 3. Search Results

From the 60 studies included in this review, a total of 35 studies investigated the efficacy of L-carnitine and CoQ10 administration for the management of fatigue symptoms, accounting for 28% and 27% of the studies, respectively. Vitamins C and D are the next most studied nutrients, accounting for 12% of the studies each. The remaining 21% of the studies investigated different nutrients including NADH, thiamine, methionine, zinc, and Myers’ cocktail (Figure 2A). Furthermore, it can be seen from (Figure 2B) that most of the included studies investigated oral administration of nutrients, with only 10 and 3 of the studies employing intravenous (IV) and intramuscular (IM) administration routes, respectively. As depicted in (Figure 2C), studies were also sorted based on the effects of each nutrient on measures of fatigue. In most cases studies reported statistically significant positive outcomes after a period of nutrient therapy. Nevertheless, several studies investigating nutrients such as CoQ10, L-carnitine, vitamin C, and vitamin D, also reported no significant improvements in fatigue symptoms. In terms of statistically significant negative outcomes, only one study reported worsening of symptoms as a result of L-carnitine administration in cancer patients undergoing chemotherapy [13].

Differentiating the studies with regards to the population they were studied in, revealed that cancer [14], CFS [15], fibromyalgia [16], and MS patients [17] are the most studied populations (Figure 3). The remaining studies investigated the effects of nutrient therapy in the elderly and healthy subjects and those suffering from other medical conditions such as Metabolic syndrome (MetS) [18], end-stage heart failure [19], kidney transplant recipients (KTRs) [20], post-stroke fatigue (PSF) patients with vitamin D deficiency [21], etc. Additionally, it can be postulated from (Figure 3) that L-carnitine [22] and vitamin C [14] are the most employed nutrients for the management of fatigue symptoms in cancer patients. Similarly, CoQ10 is amongst the most administered nutrients in FM patients [16] as well as healthy subjects [23]. Patients with CFS might also benefit from CoQ10 administration [15].

## 4. Oral Supplementation

### 4.1. Clinical Populations

#### 4.1.1. Vitamins and Minerals

Several studies have suggested the effects of oral supplementation with vitamins and minerals on fatigue amongst individuals suffering from different medical conditions (Table 1) [17]. Fatigue is the most common symptom of Multiple Sclerosis (MS), affecting up to 90% of those with MS, and is often reported as the first symptom noted by patients prior to diagnosis [7]. Moreover, fatigue is suggested to be more severe and disabling amongst MS patients than in healthy controls and others with chronic illness [7]. The administration of thiamine in MS patients for the management of fatigue has been investigated by Sevim et al. [24]. Daily oral administration of 400 mg sulbutiamine, a synthetic compound which constitutes two thiamine molecules, for two months was found to be effective in treating fatigue in MS. However, the effect was only observed in patients who were on disease modifying treatment, but not on those who were not [24]. Moreover, the effect of Alfacalcidol (1 a-hydroxycholecalciferol), a synthetic analogue of vitamin D, on MS-related fatigue has been investigated [25]. Intervention group received vitamin D3 alfacalcidol (1 μg or 40 IU) daily. Alfacalcidol lowered the mean Fatigue Impact Scale (FIS) score as compared with placebo and improved quality of life in patients with MS [25]. Moreover, Bager et al. has demonstrated the significant beneficial effect of high-dose oral thiamine (600–1800 mg/d) on chronic fatigue in patients with inflammatory bowel disease (IBD), following 4 weeks of treatment [26].

Vitamin D administration has been also reported to be associated with improvement of fatigue in various populations. Weekly administration of 50,000 IU oral cholecalciferol in patients with juvenile-onset systemic lupus erythematosus (SLE) improved many aspects of fatigue, including measures of ‘fatigue easily’, ‘fatigue during exercise’, ‘fatigue to medium efforts’, and ‘fatigue considered a problem’ [58]. Furthermore, oxidative stress is thought to underlie fatigue, with serum markers of oxidative stress being associated with symptoms of CFS [14]. Since Vitamin C is a well-known antioxidant, vitamin C treatment for the management of fatigue symptoms has been investigated. Namely, in obese adults on a weight loss program with exercise, 500mg/day of oral vitamin C for four weeks was reported to be better than placebo in reducing the rate of perceived exertion and lowering fatigue scores [14]. The majority of studies, however, have investigated the effects of intravenous vitamin C on fatigue, which is furthered elaborated in Section 5.

#### 4.1.2. Co-Enzymes

CoQ10 and NADH, common dietary supplements with purported cardioprotective effects, have been shown to significantly relieve fatigue symptoms [5]. Emerging data suggest that CFS and fibromyalgia are associated with deficiencies of CoQ10 and NADH, both of which play a pivotal role in mitochondrial ATP production and cellular metabolism homeostasis [5]. While mitochondrial failure decreases the rate of ATP synthesis which is the main agent of energy production in CFS, CoQ10 and NADH enhance cellular ATP production via mitochondrial oxidative phosphorylation [5]. Accordingly, in a study by Castro-Marrero et al., a clinical improvement in fatigue symptoms was demonstrated following initiation of oral NADH or CoQ10 supplementation in patients with CFS [29]. Thus, CoQ10 (200 mg/day) plus NADH (20 mg/day) administration is potentially a safe therapeutic approach for minimizing perceived cognitive fatigue and enhancing the health-related quality of life of individuals with ME/CFS [28]. Moreover, administration of NADH has been shown to be effective in the management of CFS symptoms [61]. In a study by Forsyth et al. [60], 10 mg of NADH was administered in CFS patients for a 4-week period. In this pilot study NADH was suggested as a valuable adjunctive therapy in the management of CFS [60]. However, it should be noted that another RCT study on the effects of 150 mg/day CoQ10 treatment in patients with CSF for two months failed to reveal any significant improvements (*p* > 0.05) in fatigue symptoms [15].

Oxidative stress and mitochondrial dysfunction have been found to play an important role in the pathogenesis of FM [62]. Furthermore, antioxidant proteins including catalase and superoxide dismutase (SOD) have been found to be diminished in FM [62]. While CoQ10 plays an important role in mitochondrial ATP production and cellular metabolism, fibromyalgia has been described to be linked with CoQ10 deficiency [62]. Accordingly, Miyamae et al. demonstrated that Ubiquinol-10 treatment in patients with juvenile FM and CoQ10 deficiency improved chronic fatigue scores as measured by the CFQ 11 [33]. Therefore, CoQ10 administration may cause remarkable improvements in FM patients. Cordero et al. investigated the efficacy of CoQ10 treatment (300 mg/day) on clinical and molecular parameters in fibromyalgia [16]. Forty days of CoQ10 administration resulted in significant reductions in pain, fatigue, and morning tiredness subscales of FIQ. Additionally, gene expression of IL-6, IL-8, and TNF was significantly reduced [16]. A higher dose of oral CoQ10, administered at 200 mg twice a day for three months, also seemed to statistically relieve fatigue symptoms in FM patients by approximately 22% [34].

Ultimately, while MS is a chronic inflammatory disorder accompanied by fatigue and depression [63], CoQ10 has neuroprotective and antioxidant properties, and hence decreases pro-inflammatory cytokines and protects the brain cells and neurons against central neurotoxic damages [64]. Therefore, the effect of CoQ10 administration on fatigue in patients with MS has been investigated [32]. The study reported a greater reduction in fatigue on the fatigue severity scale in the CoQ10 group (500 mg/day for 12 weeks) than the placebo group [32]. Overall, CoQ10 supplementation has been investigated for the improvement of fatigue in various medical conditions. Three-month administration of 60 mg/day of CoQ10 in patients with end-stage heart failure awaiting heart transplantation caused a significant reduction in fatigue during activities of daily living, in addition to significant improvements in nocturia and dyspnoea [19]. Furthermore, in an RCT study by Peel et al., the efficacy of CoQ10 (100 mg/day for 2 moths) in alleviating fatigue symptoms in late-onset sequelae of poliomyelitis was investigated [37]. The results of the study, however, did not indicate any statistically significant (*p* > 0.05) reduction in fatigue [37]. Similarly, Lesser et al. did not support the efficacy of CoQ10 administration (300 mg/day for three months) in fatigue reduction in newly diagnosed patients with breast cancer [38]. Therefore, there is a need for further research on the effect of CoQ10 on fatigue symptoms amongst different populations and employing different doses and treatment durations.

#### 4.1.3. Amino Acids

Carnitine, derived from the methylation of the amino acid lysine, plays an important role in the metabolism of fatty acids as the control of fatty acid oxidation is vested in the carnitine palmitoyltransferase system [65]. Moreover, skeletal and cardiac muscles, expressing carnitine palmitoyltransferase I (CPT I), use fatty acids as their primary source of energy. Therefore, in general, carnitine deficiency is associated with low energy levels, muscle weakness and general fatigue [65]. Cancer-related fatigue, characterized by a persistent sense of severe physical and psychological exhaustion related to cancer or its treatment, is amongst the most common symptoms in cancer patients [66]. Branched-chain amino acids have been suggested to reduce central fatigue [66]. Accordingly, Iwase et al. investigated the efficacy of a supplement containing branched-chain amino acids (2500 mg), CoQ10 (30 mg), and L-carnitine (50 mg) in the management of fatigue in breast cancer patients [22]. The significant reduction in fatigue scores suggested that the investigated intervention may be useful in controlling moderate-to-severe cancer-related fatigue [22]. However, Hershman et al. reported that in breast cancer patients undergoing adjuvant taxane-based chemotherapy, 3 g/day of oral L-carnitine for 24 weeks did not result in any significant changes in fatigue measures [13]. Another important finding of the study is that the results of the trial suggested a detrimental effect of the ALC intervention on chemotherapy-induced peripheral neuropathy (CIPN) [13]. Clearly, the use of nutritional supplements should be discouraged when there is evidence of adverse effects on any of the symptoms of the condition. Proven efficacy on different aspects of the condition should be available before any administration to avoid any potential harm. Nevertheless, chemotherapy medications including Ifosfamide and cisplatin cause urinary loss of carnitine; hence, carnitine treatment has been suggested for restoration of the carnitine pool and improving the chemotherapy-induced fatigue. Namely, administration of 4 g oral levocarnitine daily for 7 days was shown to ameliorate chemotherapy-induced fatigue in cancer patients [45]. Gramignano et al. also demonstrated that administration of 6 g/day of L-carnitine in cancer patients, significantly improved fatigue scores [46]. Altogether, there is a need for further studies investigating the effects of L-carnitine administration in patients with cancer and undergoing different treatments to ensure effective and safe administration of L-carnitine in this population.

L-carnitine administration has been investigated for the management of fatigue-related symptoms in several different conditions. Fatigue-related symptoms in hypothyroid patients have been suggested to be related to the relative deficiency of carnitine in these patients. Thyroid hormone plays an essential role in carnitine-dependent fatty acid import and oxidation and decreased carnitine levels in hypothyroidism may be explained by decreased biosynthesis of carnitine [39]. Therefore, An et al. investigated the effects of L-carnitine treatment on fatigue-related symptoms in hypothyroid patients [39]. It was demonstrated that administration of L-carnitine (990 mg L-carnitine twice daily) in hypothyroid patients significantly improved physical fatigue score (PFS) and mental fatigue score (MFS) in patients younger than 50 years and those with free T3 ≥ 4.0 pg/mL [39]. Furthermore, levocarnitine tartrate administration (1000 mg daily for 12 weeks) has been found to significantly improve muscle weakness and fatigue in children with neurofibromatosis type 1 (NF1) [47].

Lastly, S-adenosylmethionine (SAM) is a methyl donor with a critical role in many metabolic processes. SAMe exerts anti-inflammatory, antidepressant, and analgesic effects, and is suggested to have tolerability equal to or better than the non-steroidal anti- inflammatory drugs [53]. The efficacy of 800 mg orally administered SAM daily versus placebo for six weeks was investigated in FM patients [53]. SAM treatment resulted in significant improvements in FM patients with regards to fatigue, clinical disease activity, morning stiffness, pain, and mood symptoms [53].

### 4.2. Non-Clinical Populations

#### 4.2.1. Vitamins and Minerals

Administration of vitamins and minerals has also been investigated in populations without any known medical condition. Zinc is an intracellular signalling molecule which plays a critical role in various physiological processes including cellular proliferation, DNA repair, anti-inflammatory responses, immune system regulation, adenosine triphosphate (ATP) functioning, and regulation of enzymatic and muscle function [67]. Furthermore, zinc is vital for the control of proliferation, differentiation, and programmed cell death [68]. Namely, chronic zinc deprivation is associated with accelerated proliferation of vascular smooth muscle cells, which, in combination with calcification, can aggravate the progression of atherosclerosis [69]. Serum zinc concentration also diminishes with aging, with about 35% to 45% of the elderly having zinc levels lower than the normal range [70]. Regarding supplementation, in a study conducted on 150 elderly subjects aged ≥60 years, daily administration of 30 mg of zinc for 70 days significantly reduced fatigue and increased serum zinc levels [54]. Moreover, in a randomized, double-blind, placebo-controlled trial, administration of 220 mg of zinc sulphate in women with premenstrual syndrome (PMS) from the 16th day of each menstrual cycle to the 2nd day of the next for 3 months, resulted in significant improvements in fatigue scores as well as other symptoms of PMS monitored using the premenstrual symptoms screening tool [55]. Moreover, this improvement tended to increase each month, potentially due to the gradual improvement of zinc status [55].

#### 4.2.2. Co-Enzymes

Several studies have investigated the administration of CoQ10 for improvement of fatigue in non-clinical populations [23], where the efficacy of CoQ10 administration on physical fatigue was examined using physical workload trials. Administration of 100 or 150 mg/day ubiquinol-10, the reduced form of CoQ10, was investigated by Mizuno et al. [23]. Subjective levels of fatigue sensation and sleepiness after cognitive tasks improved significantly in both groups compared with those in the placebo group. Additionally, the group supplemented with 150 mg/day of ubiquinol-10 showed significant improvements compared with the control group in parameters such as serum level of oxidative stress, subjective level of relaxation after task, sleepiness before and after task, as well as motivation for task [23]. Moreover, in a study by the same group, 300 mg, but not 100 mg of CoQ10 administration alleviated the recovery period and the subjective fatigue sensation measured on a visual analogue scale [30]. In another study on the effects of CoQ10 administration on exercise performance in soccer players, four weeks of 300 mg/day CoQ10 administration did not result in any significant changes in fatigue scores as well as weight and body fat percent [36]. However, VO_2_ max and performance in soccer players were significantly improved [36]. Gokbel et al. also investigated the efficacy of supplementation with 100 mg/day of CoQ10 on performance during repeated bouts of supramaximal exercise in sedentary men [31]. During the study period, five Wingate tests (WTs) were performed at baseline and after CoQ10, or placebo administration. Although CoQ10 resulted in a significant increase in mean power during the WT5, the observed decreases in fatigue indexes following 100 mg CoQ10 administration did not differ from that seen with placebo administration [31].

Several studies also reported no significant reduction in fatigue outcomes following CoQ10 administration [35]. Namely, the results from a study on the effects of CoQ10 on fatigue in obese subjects failed to show any significant change in mean FSS score between the placebo and CoQ10 groups [35]. The results of this study might be affected by the small sample of size of the trial [35]. Nonetheless, further studies on a larger sample size are required since changes in subjective fatigue between groups were not significantly different, even though the fatigue level improved significantly in the CoQ10 group.

#### 4.2.3. Amino Acids

L-carnitine may be effective in improving cognitive status and physical functions in the elderly. L-carnitine administration has been found to reduce both mental and physical fatigue in aged subjects [48]. Malaguarnera et al. demonstrated that L-carnitine administration in the elderly (2 g twice a day) resulted in significant improvements in physical and mental fatigue, severity of fatigue, functional status, cognitive functions, muscle pain and sleep disorders [50]. The effects of acetylcarnitine and propionylcarnitine on the symptoms of CFS have been compared. It has been suggested that while Acetylcarnitine had a significant effect on mental fatigue and propionylcarnitine on general fatigue, both treatments improved attention concentration. However, less improvement was found for the combined treatment [51]. L-carnitine was also compared to androgen in the treatment of male aging symptoms. Subjects were given testosterone undecanoate 160 mg/day or propionyl-L-carnitine 2 g/day plus acetyl-L-carnitine 2 g/day. Both treatments significantly diminished the fatigue scale score at 3 months, and showed significant results for treatment of male aging symptoms [52].

### 4.3. Nutrient Deficiencies 

#### 4.3.1. Vitamins and Minerals

Vitamin deficiency is prevalent and has been found to be associated with fatigue in different populations. Accordingly, several studies have investigated vitamin D administration for the management of fatigue in subjects with vitamin D deficiency or insufficiency (i.e., suboptimal levels which are not low enough to be classified as deficient). While normal vitamin D levels typically range from 30 to 100 ng/mL, insufficient vitamin D levels are defined by serum levels between 20 ng/mL and 29 ng/mL, and serum levels below 20 ng/mL are classified as vitamin D deficiency [58]. Roy et al. [56] reported that the prevalence of low vitamin D was 77.2% in patients who presented with fatigue. Normalization of vitamin D levels by ergocalciferol (Vitamin D2) therapy for five weeks resulted in significant improvement in fatigue scores (*p* < 0.001) in all five subscale categories of FAS questionnaire [56]. Similarly, Nowak et al. reported that vitamin D administration in individuals presenting fatigue and vitamin D deficiency significantly improved FAS scores, with the improvements correlating with the rise in 25(OH)D levels [57]. Han et al. also demonstrated that serum 25(OH)D levels were inversely and independently related to fatigue scores in kidney transplant recipients (KTRs) exhibiting vitamin D deficiency [20]. Moreover, it was indicated that while fatigue was found in 40.1% of KTRs, vitamin D3 administration significantly increased 25(OH)D levels and improved fatigue symptoms in these patients [20]. Furthermore, vitamin D administration has been investigated for the management of post-stroke fatigue in patients with primary acute ischemic stroke (AIS) and vitamin D deficiency [21]. The study reported significant reduction in FFS scores in the study group compared to the control group, at both one month (t = −4.731, *p* < 0.01) and three months (t = −7.937, *p* < 0.01) following vitamin D administration [21]. Lastly, 12 weeks of treatment with 50,000 IU vitamin D3 weekly in post-menopausal women with early-stage breast cancer exhibiting vitamin D deficiency or insufficiency was investigated [59]. However, the difference between the fatigue scores of subjects exhibiting 25OHD levels above the median (66 ng/mL) and those with 25OHD levels below the median were not statistically significant. Overall, vitamin D deficiency co-presents in many medical conditions in association with fatigue symptoms.

Additionally, it has been indicated that the fatigue and related manifestations concomitant with MS are associated with an intracellular mild thiamine deficiency [17]. Costantini et al. demonstrated that high-dose thiamine therapy (600–1500 mg/day orally or 100 mg/mL once a week parenterally) was effective in reversing fatigue in MS [17]. Interestingly, it was demonstrated that improvement in fatigue was observed within hours from the first parenteral administration or within 2–3 days following initiation of the oral therapy [17].

#### 4.3.2. Amino Acids

As mentioned, carnitine deficiency can result in low energy levels, muscle weakness and general fatigue [65]. In cancer patients, carnitine deficiency is amongst the many metabolic disturbances that may contribute to fatigue. L-carnitine administration (1500 mg/day of levocarnitine per os) has been shown to improve general fatigue in cancer patients during chemotherapy [40]. A few studies by Cruciani et al. have explored administration of L-carnitine in cancer patients with L-carnitine deficiency [41]. In these studies, carnitine deficiency was defined as free carnitine < 35 mM/L for males or <25 mM/L for females, or an acyl-carnitine ratio (total carnitine minus free carnitine/free carnitine) > 0.4 [41]. Thereafter, cancer patients with carnitine deficiency were assigned to successive dose groups, starting at 250 mg/day and increasing in each group by 500 mg/day to a maximum dose target of 3000 mg/day [41]. The results showed a significant decrease in measures of fatigue (Brief Fatigue Inventory, BFI) with a dose-response relationship for free-carnitine levels and fatigue (BFI) scores, suggesting that L-carnitine may be safely administered at doses up to 3000 mg/day [71]. However, a couple of investigations by Cruciani et al. failed to show any significant improvements in fatigue symptoms with L-carnitine treatment [43]. In the study investigating the effects of L-carnitine supplementation as a treatment for fatigue in patients with cancer, four weeks of 2 g/day of L-carnitine administration failed to improve fatigue in patients with invasive malignancies [43]. However, the reported results might be due to the dose and duration of L-carnitine administration employed in this study, which are different from those of some other studies showing positive outcomes. Furthermore, no significant improvement in fatigue symptoms was observed in terminally ill HIV/AIDS patients with carnitine deficiency and fatigue receiving 3 g/day of oral L-carnitine for 2 weeks [44]. It should be noted that this study might have been less representative due to several factors such as poor participant accrual, the excessive number of outcome measures, and effect size of the study [44].

## 5. Parenteral Administration

### 5.1. Clinical Populations

Parenteral administration (intravenous or intramuscular) enables high plasma concentrations that are not achievable through oral administration [72]. For example, it has been reported that oral administration of vitamin C at a dose of 1.25 g daily until participants achieved a steady state for this dose led to maximum plasma concentration of 134.8 ± 20.6 μmol/L, while IV administration of vitamin C at the same dose resulted in a maximum plasma concentration of 885 ± 201.2 μmol/L [73]. This is because intravenous administration bypasses the intestinal absorption system, thus allowing plasma concentrations to be elevated to concentrations that are unachievable via oral administration [74]. Several studies have investigated parenteral administration routes for administration of micronutrients (Table 2). Studies have suggested the efficacy of IVC on improving the quality of life (QoL) of cancer patients by improving fatigue symptoms and reducing the toxic side effects of chemotherapy [75]. In a multicentre, open-label, observational study investigating the effects of IVC on the quality of life of cancer patients, significant decreases were observed in fatigue scores following four weeks of IVC therapy [76]. Similarly, Yeom et al. investigated the impact of intravenous vitamin C on the quality of life of cancer patients in an observational study [77]. Intravenous administration of 10 g vitamin C twice with a 3-day interval and an oral intake of 4 g vitamin C daily for a week resulted in significantly lower scores of fatigue in the studied patients [77]. Similar results regarding the impacts of IVC on fatigue scores were observed in an epidemiological, retrospective cohort study with parallel groups in which breast cancer patients were treated with 7.5 g of IVC in addition to their standard tumour therapy for at least 4 weeks [78].

Intravenous nutrient therapy (IVNT), using a modified Myers’ intravenous nutrient formula, has been evaluated for the management of pain levels, fatigue, and activities of daily living in FM patients who had failed numerous medical therapies such as nonsteroidal anti-inflammatory drugs (NSAIDs) and occasional opioid medications for pain control, and had very poor quality of life secondary to pain and fatigue. The modified Myers’ intravenous nutrient formula contained magnesium chloride hexahydrate, Calcium gluconate, Vitamin C, Hydroxocobalamin (B12), Pyridoxine hydrochloride (B6), Dexpanthenol (B5), Riboflavin (B2), Thiamine (B1), and Niacinamide. Administration of IVNT in therapy-resistant FM patients resulted in increased energy and improved activities in daily living as well as significant decreases in pain (60%) and fatigue (80%) with no side effects reported. However, no participants reported complete resolution of pain and fatigue [84]. Intramuscular S-adenosyl-L-methionine (SAMe) has been also investigated and compared to transcutaneous electrical nerve stimulation (TENS) for the management of fibromyalgia. Unlike TENS, daily administration of 200 mg intramuscular SAMe plus 200 mg tablets for 7 days was found to significantly reduce subjective symptoms of pain and fatigue in FM patients [85].

As well as combination nutrients, the influence of specific amino acids on fatigue has been investigated. As previously mentioned, L-carnitine plays an important role in lipid metabolism as it promotes the transportation of long-chain-fatty-acid across the mitochondrial membrane [65]. This facilitates the cellular breakdown and energy liberation of stored fat reserves [86]. Intravenous l-carnitine administration has been investigated in patients with metabolic syndrome (MetS) [18]. The treatment group received 4 g/day of intravenous L-carnitine for 7 days, while patients in the CT group were injected with saline. It was demonstrated that L-carnitine administration facilitated fasting-induced weight loss in MetS patients in the LC group compared to the control group. Moreover, physical fatigue and fatigue severity were significantly reduced in the LC group, but were aggravated in the control group [18].

### 5.2. Non-Clinical Populations

The effect of intravenous vitamin C (IVC) on fatigue has been evaluated amongst office workers [79]. Early indications of vitamin C deficiency may be manifested as fatigue, malaise, and a reduced desire to be physically active [79]. The results of the study suggested that the fatigue scores decreased significantly in the vitamin C group after two hours and remained lower for one day. It should also be noted that the effect of the intervention was strongest in subjects with lower baseline levels of vitamin C [79].

### 5.3. Nutrient Deficiencies

Intravenous supplementation of micronutrients has been investigated in patients exhibiting deficient/insufficient levels of different micronutrients. Vitamin D deficiency has been found to be associated with non-specific musculoskeletal pain, headache, and fatigue. The liver plays an important role in metabolism of vitamin D, and diseases of the liver interfere with production of the active metabolites of vitamin D. The effect of vitamin D administration on fatigue presence and severity in patients with liver cirrhosis due to chronic hepatitis has been investigated in a study by Aziz et al. [82]. The results of this study indicated that administration of a single dose of vitamin D3 significantly improved fatigue in vitamin D-deficient liver cirrhosis patients [82]. Moreover, in patients with end-stage renal disease (ESRD), L-carnitine insufficiency indicated by a ratio of plasma acylcarnitine to carnitine concentration greater then 0.4, has been found to be associated with impaired functional capacities. In the investigated ESRD patients, intravenous administration of L-carnitine at the conclusion of each thrice-weekly dialysis session for 24 weeks, significantly improved the fatigue domain of the Kidney Disease Questionnaire (KDQ) [83]. 

Ultimately, clinical vitamin C deficiency, defined as a plasma concentration of <0.2 mg/dL, is common in inflammatory diseases [80]. Accordingly, vitamin C has been suggested to be beneficial in the management of allergies. Oxidative stress is a key factor in the pathogenesis of allergic disease [87]. Furthermore, allergies are associated with reduced plasma levels of ascorbate, which play a role in preventing excessive inflammation without reducing the defensive capacity of the immune system [80]. Therefore, vitamin C administration has been investigated for the management of allergic diseases. Vollbracht et al. [80] investigated the effects of IVC on disease nonspecific symptoms of patients with allergy-related respiratory or cutaneous indications, representing vitamin C deficiency. Adjuvant treatment with IVC significantly decreased fatigue symptoms in 93.5% of patients [80]. Lastly, the effect of IVC on opiate consumption and pain in patients undergoing laparoscopic colectomy and exhibiting plasma vitamin C concentrations of <20 μmol/L has been investigated [81]. While IVC was reported to significantly reduce pain scores at rest during first 24 h postoperatively, it failed to make any significant changes in fatigue score 2, 6, and 24 h post operation [81].

## 6. Discussion

The impact of fatigue on quality of life is substantial and common in several different diseases, including multiple sclerosis, cancer, metabolic syndrome, fibromyalgia and myalgic encephalomyelitis. Furthermore, a significant number of individuals with fatigue do not have a confirmed underlying diagnosis. Managing fatigue within a medical setting is challenging, with no specific treatment available outside of supportive therapies, which have mixed efficacy and show large inter-individual variability. Therefore, there is an impetus for exploring novel treatments to limit fatigue in both clinical and non-clinical populations. Nutrients play an essential role in a variety of basic metabolic pathways that support fundamental cellular functions affecting mental and physical fatigue. Energy production and metabolism is complex, with many nutrients and enzymes playing important roles in its preservation. Therefore, combining the nutrients of interest together as one intervention could potentially provide an effective option for the management of fatigue associated with chronic diseases.

The current review explores existing evidence for the efficacy of several nutrients and their potential benefit as both a stand-alone treatment for fatigue and as adjunct to other interventions. Therefore, literature was reviewed covering fatigue in both apparently normal physiological states, as well as several disease states commonly associated with fatigue. Oral supplementation with different nutrients has been explored with several studies revealing statistically significant benefits to fatigue outcomes. These include CoQ10, L-carnitine, zinc, methionine, NADH, and vitamins C, D, and B. CoQ10 and L-carnitine have been shown to improve fatigue in several conditions when administered orally or parenterally in case of L-Carnitine. Therefore, further research should be conducted to explore these nutrients, in combination or as stand-alone treatments, and optimal doses, most efficacious routes of administration, and safety profiles across various clinical and non-clinical populations need to be identified. High-dose IVC has shown promise for reducing fatigue associated with several different disease pathologies. Further comparisons between injectable and oral administrations of vitamin C may also be helpful to determine the role this vitamin plays in the management of fatigue and fatigue related conditions. It is apparent that several other nutrients are of interest in the management of fatigue. Studies have typically focused on single nutrient approaches, with a multi-nutrient approach having been assessed in a single study, using a modified Myers’ intravenous nutrient formula in a small group of treatment resistant individuals with fibromyalgia. Therefore, multi-nutrient treatments, administered parentally, should be further explored on non-treatment resistant patients with fibromyalgia and other fatigue-based illness cases.

While fatigue is not only experienced by those with clinical conditions, with a third of people with fatigue having no known aetiology, it should be kept in mind that treatments which might be helpful for the healthy population may differ from those with disease-related fatigue. From the studies reported in the current review, five studies assessed perceived fatigue in healthy volunteers, with significant reduction in fatigue with different oral and parenteral single nutrient interventions including L-carnitine, zinc, CoQ10 and vitamin C. This may support the concept that nutrient therapy could be beneficial for managing fatigue both in patients with disease and illness and those perceived as healthy. Therefore, the underlying aetiology of fatigue symptoms needs to be investigated in healthy individuals as well as those with known physiological conditions. Moreover, the associations between various nutrients and different clinical manifestations such as fatigue need to be established, as it allows identifying potential treatments and replacement therapies. Nutrient therapy might also be helpful in the management of fatigue symptoms associated with viral conditions including COVID-19 infection and its prolonged illness, long COVID; both often present with fatigue. Moreover, several studies have investigated the role of nutrients such as vitamin D in the treatment of sarcopenia, due to the positive impact of vitamin D supplementation on muscle mass and function [88]. However, the impact of nutrient supplementation on fatigue symptoms amongst individuals suffering from sarcopenia has not been investigated. Therefore, nutrients which have been found to be associated with improvement of muscle function including vitamin D and creatines [89], should also be specifically investigated for their potential impacts on fatigue symptoms.

One challenge, when exploring interventions for improving fatigue is how this symptom is evaluated. Most commonly the data are collected either through an accepted scoring methodology, such as FAS and CFQ 11, or using an individual’s subjective state. Therefore, there are notable methodological differences across studies, making direct and between-study comparisons difficult. Moreover, when evaluating the effect of vitamin administration in clinical trials, subgroup analysis of men and women should be performed as the effect may differ between the sexes. Accordingly, a study by Klasson et al. demonstrated that while there was a significant correlation between 25-OHD and fatigue scores in men, with higher 25-OHD levels associated with less fatigue, no correlation between 25-OHD and fatigue was seen for women [90]. Moreover, a study by Khan et al. explored the effect of vitamin D treatment on serum 25-hydroxy vitamin D levels and fatigue in post-menopausal women with breast cancer [59]. Even though those who exhibited 25OHD levels above the median (66 ng/mL) reported lower BFI scores (median = 1.4) than women with 25OHD levels below the median (BFI median = 2.9), the difference between the two groups was not statistically significant [59]. Lastly, when considering a nutrient therapy for the management of disease-related fatigue symptoms, the impact of the intervention on other aspects of the clinical condition must be carefully evaluated to avoid any adverse effects or worsening of any other symptoms. Namely, in a study by Hershman et al., it was noticed that L-carnitine administration in cancer patients undergoing chemotherapy increased chemotherapy-induced peripheral neuropathy (CIPN) [13]. Therefore, randomized controlled trials (RCTs) are crucial to validate the potential beneficial impact of supplementation with different nutrients and in different populations. These studies will ensure avoiding nutritional supplements that do not have proven efficacy or might have potential harm.

## 7. Conclusions

Altogether, it is well demonstrated that nutrient therapy might be beneficial for management of fatigue symptoms in both healthy and clinical populations. Several nutrients have shown good efficacy and safety at different doses, routes, and frequency of administration in humans. However, further research into nutritional interventions to treat or reduce fatigue should be conducted investing a broader range of nutrients. Randomised, placebo-controlled trials should compare existing interventions with these novel approaches, testing both individual and multi-nutrient approaches, in different groups experiencing fatigue. These interventions could also be tested as adjuncts to existing interventions, to explore enhanced efficacy. If found efficacious and replicable, then new nutraceuticals may be developed to manage this debilitating symptom.

## Figures and Tables

**Figure 1 nutrients-15-02154-f001:**
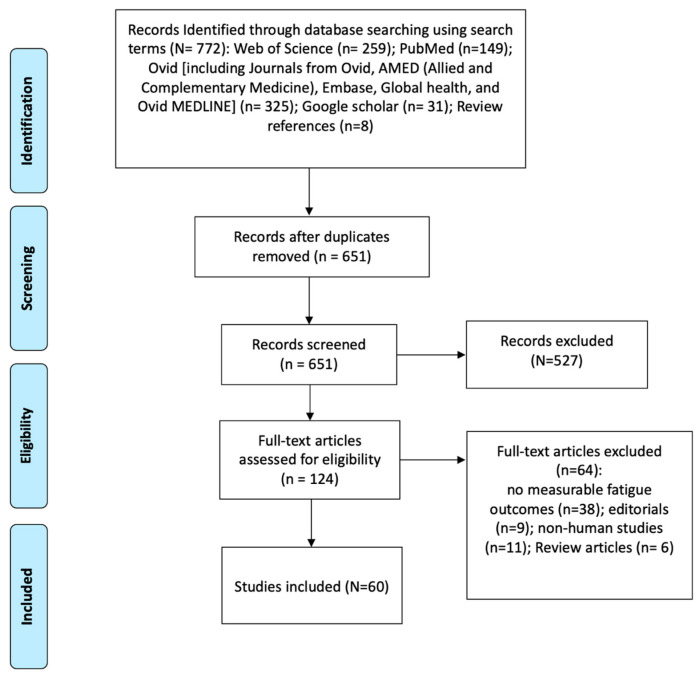
Diagram of the methodology used for the literature review process.

**Figure 2 nutrients-15-02154-f002:**
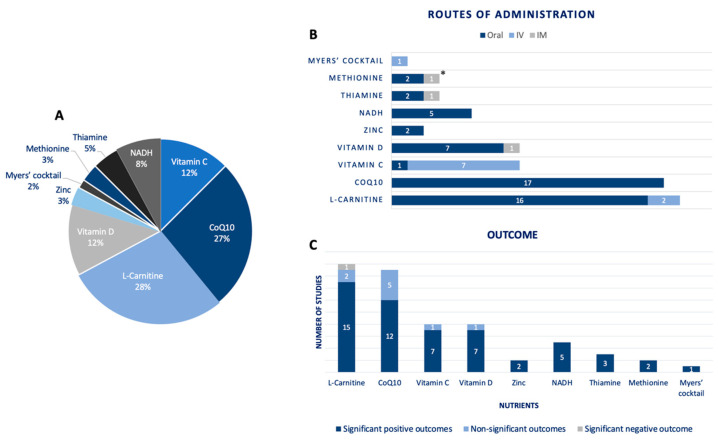
(**A**) Percentage of studies investigating the effects of each nutrient for the treatment of fatigue symptoms. (**B**) Investigated nutrients based on their route of administration. The plot demonstrates number of studies employing oral, intravenous, or intramuscular routes of administration for each nutrient. (* both oral and IM administration employed by one of the identified studies) (**C**) Investigated nutrients based on their effects on fatigue outcome measures, plotted as number of studies reporting significant positive outcome, significant negative outcome, or non-significant outcomes.

**Figure 3 nutrients-15-02154-f003:**
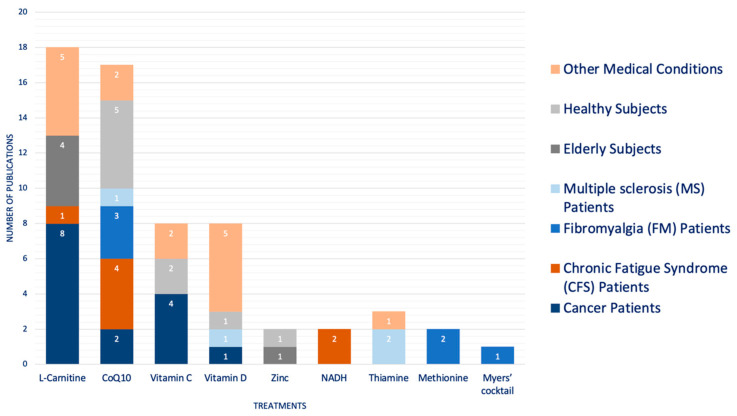
Studied populations for investigating the effects of each nutrient on fatigue symptoms. Number of studies investigating each nutrient for the management of fatigue in different clinical and non-clinical populations are demonstrated.

**Table 1 nutrients-15-02154-t001:** Studies employing oral supplementation.

Reference	Study Design	Group	Treatment	No. of Participants	Main Findings
**Coenzyme Q10**					
Castro-Marrero et al., 2015 [27]	Randomized, double-blind placebo-controlled trial	Chronic fatigue syndrome (CFS) patients	Oral CoQ10 (200 mg/day) + NADH (20 mg/day) for 8 weeks	73	Significant improvement of fatigue showing a reduction in fatigue impact scale total score (*p* < 0.05)
Castro-Marrero et al., 2021 [28]	Prospective, randomized, double-blind, placebo-controlled trial	Individuals with Myalgic encephalomyelitis/chronic fatigue syndrome (ME/CFS)	Oral CoQ10 (200 mg/day) + NADH (20 mg/day) for 12 weeks	207	Significant reduction in cognitive fatigue perception and overall fatigue impact scale (FIS-40) score (*p* < 0.001 and *p* = 0.022, respectively)
Castro-Marrero et al., 2016 [29]	Randomized, controlled, double-blind trial	Chronic fatigue syndrome (CFS) patients	50 mg of CoQ10 and 5 mg of NADH twice daily for 8 weeks	80	Significant reduction in perception of fatigue through all follow-up visits in active group versus placebo (*p* = 0.03)
Iwase et al., 2016 [22]	Multi-institutional, randomized, exploratory trial	Breast cancer patients	Inner Power (IP) oral supplement containing branched-chain amino acids (2500 mg), coenzyme Q10 (30 mg), and L-carnitine (50 mg), once daily for 21 days	57	Changes in the worst level of fatigue, global fatigue score (GFS), and current feeling of fatigue were significantly different between the intervention and control groups
Mizuno et al., 2008 [30]	Double-blinded, placebo-controlled, three crossover design	Healthy volunteers	Oral coenzyme Q10 (100 or 300 mg/d) or placebo administration for 8 days	17	Significant alleviation of subjective fatigue sensation measured on a visual analogue scale in the 300-mg coenzyme Q10–administered group after
Mizuno et al., 2020 [23]	Double-blind, placebo-controlled study	Healthy volunteers	Ubiquinol-10 (100 or 150 mg/d) or placebo administration for 12 weeks	104	Improvements in subjective levels of fatigue sensation and sleepiness after cognitive tasks
Gokbel et al., 2010 [31]	Randomized, double-blind, crossover study	Healthy and sedentary men	100 mg/day CoQ10 for two 8-week periods	15	Mean power increased only with CoQ10 supplementation during the fifth Wingate test (WT5)
Sanoobar et al., 2016 [32]	Randomized, double-blinded, placebo-controlled trial	Multiple sclerosis patients	500 mg/day CoQ10 for 12 weeks	48	Significant decrease of fatigue severity scale (FSS) was observed in CoQ10 group during the intervention (*p* = 0.001)
Cordero et al., 2013 [16]	Randomized, double-blind, placebo- controlled trial	Fibromyalgia (FM) patients	CoQ10 supplementation (300 mg/day) for 40 days	20	Prominent reduction in pain (*p* < 0.001), fatigue, and morning tiredness (*p* < 0.01)
Miyamae et al., 2013 [33]	Double-blind, placebo-controlled trial	Patients with juvenile FM (n = 10) and healthy control subjects (n = 67)	Ubiquinol-10 (100 mg/day) for 12 weeks	77	Significant improvements in chronic fatigue scores as measured by the Chalder Fatigue Scale
Pierro et al., 2017 [34]	Randomised, open-label, cross-over study	Female Fibromyalgia (FM) patients	CoQ10 200 mg × 2/day for 3 months	22	Statistically significant relieve of the fatigue symptoms
Berman et al., 2004 [19]	Randomized, Placebo-Controlled Study	Patients with end-stage heart failure	CoQ10 60 mg U/day for 3 months	32	Statistically significant relieve of the fatigue symptoms
Lee et al., 2011 [35]	Randomized, double-blind, placebo-controlled, single- centre study	Obese subjects	CoQ10 200 mg/day for 12 weeks	36	NS, mean FSS score decreased significantly from 40.1 to 33.1 in the coenzyme Q10 group (*p* = 0.017), but no significant change was seen in the placebo group (*p* = 0.464)
Gharahdaghi et al., 2013 [36]	Randomized, double-blind placebo-controlled trial	Soccer players	CoQ10 300 mg/day for 4 weeks	16	NS, fatigue index did not significantly change (*p* = 0.27), no significant changes in body composition, significant changes in VO2max and performance
Peel et al., 2015 [37]	Parallel-group, randomized, placebo-controlled trial	Patients with late-onset sequelae of poliomyelitis	CoQ10 100 mg/day for 2 months	101	NS, no significant changes in fatigue scores (*p* = 0.36)
Fukuda et al., 2016 [15]	Open-Label study, Randomized clinical trial	Patients with chronic fatigue syndrome (CFS)	CoQ10 150 mg/day for 2 months	20	NS, no significant changes in fatigue scores (*p* > 0.05)
Lesser et al., 2013 [38]	Randomized Double-Blind, Placebo-Controlled Study	Breast Cancer subjects	CoQ10 300 mg/day for 3 months	236	NS, no significant changes in fatigue scores (*p* > 0.05)
**L-Carnitine**					
An et al., 2016 [39]	Randomized, double-blind, placebo-controlled trial	Hypothyroid patients	L-carnitine (990 mg) twice daily for 12 weeks	60	Mental fatigue score (MFS) was significantly decreased, physical fatigue score (PFS) was significantly improved in patients younger than 50 years and those with free T3 ≥ 4.0 pg/mL
Matsui et al., 2018 [40]	Single-institution, non-randomized study	Cancer patients	1500 mg/day of levocarnitine per os for 8 weeks	11	Significant reduction of general fatigue
Cruciani et al., 2006, 2004 [41]	Open-label Phase I/II clinical trial	Cancer patients	L-carnitine (250, 750, 1250, 1750, 2250, 2750, 3000 mg/day), administered in two daily doses for 7 days	38	Dose-response relationship for total—(r = 0.54, *p* = 0.03), free-carnitine (r = 0.56, *p* = 0.02) levels, and fatigue (BFI) scores (r = −0.61, *p* = 0.01).
Cruciani et al., 2009 [42]	Double-Blind, Placebo-Controlled Study	Cancer patients	L-carnitine 1 g twice daily for 2 weeks	29	Significant improvement of fatigue on the FACT-An fatigue subscale (*p* < 0.03), significant improvement of FACT-An functional well-being subscale (*p* < 0.03)
Cruciani et al., 2012 [43]	Phase III, Randomized, Double-Blind, Placebo-Controlled Trial	Cancer patients	L-carnitine 2 g/d for 4 weeks	376	NS, no statistically significant differences between the placebo and treatment arms (*p* = 0.57) in fatigue symptoms evaluated by Brief Fatigue Inventory (BFI)
Cruciani et al., 2015 [44]	Double-blind, placebo-controlled pilot study	HIV/AIDS patients with carnitine deficiency and fatigue	L-carnitine 3 g/d for 4 weeks	35	NS, no statistically significant differences in fatigue symptoms evaluated by Brief Fatigue Inventory (BFI)
Graziano et al., 2002 [45]	Prospective observational study	Cancer patients	Oral levocarnitine 4 g daily for 7 days	50	Significant improvement of fatigue measured by Functional Assessment of Cancer Therapy-Fatigue score
Gramignano et al., 2006 [46]	Open-label, non-randomized study	Cancer patients	L-carnitine 6 g/d for 4 weeks	12	Fatigue, as measured by the Multidimensional Fatigue Symptom Inventory—Short Form, was significantly decreased
Vasiljevski et al., 2021 [47]	Open-label, single-arm, single centre, phase 2a clinical trial	Children with neurofibromatosis type 1 (NF1)	Levocarnitine tartrate 1000 mg/day for 12 weeks	6	53% increase in dorsiflexion strength (*p* = 0.02), mean 66% increase in plantarflexion strength (*p* = 0.03), 10% increase in long jump distance (*p* = 0.01) and 6MWT distance (*p* = 0.03)
Pistone et al., 2003 [48]	Placebo-controlled, randomised, double-blind, two-phase study	Elderly subjects	Levocarnitine 2 g/day	84	Wessely and Powell scores decreased significantly by 40% (physical fatigue) and 45% (mental fatigue)
Malaguarnera et al., 2007 [49]	Placebo-controlled, randomized, double-blind, 2-phase study	Centenarians	Levocarnitine 2 g/day	66	Significant differences in physical fatigue, mental fatigue, fatigue severity, and MMSE; significant improvements in the following markers: total fat mass, total muscle mass, plasma concentrations of total carnitine
Malaguarnera et al., 2008 [50]	Single centre, randomized, double blind, comparative clinical trial	Elderly subjects	Acetyl L-carnitine (ALC), 32 g twice-a-day	96	Decrease in physical fatigue: 6.2 (*p* < 0.001), in mental fatigue: 2.8 (*p* < 0.001), in severity of fatigue: 21.0 (*p* < 0.001) and improvements in functional status: 16.1 (*p* < 0.001) and cognitive functions: 2.7 (*p* < 0.001)
Vermeulen et al., 2004 [51]	Open Label, randomized Study	Chronic fatigue syndrome (CFS) patients	2 g/d acetyl-L-carnitine, 2 g/d propionyl-L- carnitine, and its combination for 24 weeks	90	Acetylcarnitine significantly improved mental fatigue (*p* = 0.015) and propionylcarnitine improved general fatigue (*p* = 0.004); attention concentration improved in all groups
Cavallini et al., 2004 [52]	Randomized, double-blind placebo-controlled trial	Men older than 60 years	Propionyl-L-carnitine 2 g/day plus acetyl-L-carnitine 2 g/day for 6 months	120	Significant reduction of the fatigue scale score at 3 months (*p* = 0.01), significant improvement of the nocturnal penile tumescence and International Index of Erectile Function score
Hershman et al., 2013 [13]	Randomized Double-Blind Placebo-Controlled Trial	Cancer patients	Acetyl-L-carnitine (ALC) 3 g/d for 24 weeks	409	NS, no significant changes in fatigue scores evaluated by Functional Assessment of Chronic Illness Therapy—Fatigue (FACIT-F), Grade 3 to 4 neurotoxicity was more frequent in the ALC arm
Methionine					
Jacobsen et al., 1991 [53]	Double-blind Clinical Evaluation	Fibromyalgia (FM) patients	S-adenosylmethionine 800 mg/day for 6 weeks	44	Improvements were seen for clinical disease activity (*p* = 0.04), pain experienced during the last week (*p* = 0.002), fatigue (*p* = 0.02), morning stiffness (*p* = 0.03) and mood evaluated by Face Scale (*p* = 0.006) in the actively treated group compared to placebo
**Zinc**					
Afzali et al., 2021 [54]	Randomized clinical trial	Elderly subjects aged ≥60 years	Zinc supplement 30 mg/day for 70 days	150	Significant reduction of fatigue (mean difference: −10.41 vs. 1.37, *p* < 0.001), significant increase in serum zinc level (mean difference: 14.22, vs. −0.57, *p* < 0.001)
Siahbazi et al., 2017 [55]	Double-blind randomized and placebo-controlled trial	Women with premenstrual syndrome (PMS)	Zinc sulphate 220-mg capsules (containing 50 mg elemental zinc) from the 16th day of the menstrual cycle to the second day of the next cycles	142	Significant improvements in Premenstrual Symptoms Screening Tool (PSST) component scores including fatigue, mental and physical symptoms
**Vitamin D**					
Roy et al., 2014 [56]	Prospective non-randomized therapeutic study	Patients with fatigue and stable chronic medical conditions	Vitamin D2 (Ergocalciferol 50,000 units), three times per week for 5 weeks	174	Fatigue symptom scores improved significantly (*p* < 0.001); prevalence of low vitamin D was 77.2% in patients who presented with fatigue
Nowak et al., 2016 [57]	Double-blind placebo-controlled clinical trial	Healthy persons presenting with fatigue and vitamin D deficiency (serum 25(OH)D < 20 mg/L)	Vitamin D3 (cholecalciferol), single oral dose of 100,000 units	120	Mean fatigue assessment scale (FAS) scores decreased significantly in the vitamin D group compared with placebo (*p* = 0.01)
Han et al., 2017 [20]	Observational study	Kidney transplant recipients (KTRs)	Vitamin D3 (cholecalciferol) 800 IU/d, for 9 months	60	25(OH)D was increased with 18.5% (*p* = 0.004) and subscale fatigue of the Checklist Individual Strength (CIS) scores improved with 10.0% (*p* = 0.007)
Lima et al., 2016 [58]	Randomized, double-blind, placebo-controlled trial	Juvenile-onset systemic lupus erythematosus (SLE) patients	Vitamin D3 (cholecalciferol) 50,000 IU/week for 6 months	40	Reduction in fatigue related to social life score evaluated using the Kids Fatigue Severity Scale (K-FSS) (*p* = 0.008)
Wang et al., 2021 [21]	Retrospective cohort study	Post-stroke fatigue (PSF) patients with vitamin D deficiency	Study group: vitamin D3 (cholecalciferol, 600 IU/day) for 3 months, control group: patients with vitamin D deficiency were not treated with combined vitamin D	139	Fatigue Severity Scale score was significantly lower in the study group than in the control group at 1 month (t = −4.731, *p* < 0.01) and 3 months (t = −7.937, *p* < 0.01) after treatment
Khan et al., 2010 [59]	Prospective observational study	Post-menopausal women with early-stage, receptor-positive invasive breast cancer	Vitamin D3 (cholecalciferol) 50,000 IU/week for 12 weeks	51	NS, the difference between the fatigue scores of subjects exhibiting 25OHD levels above the median (66 ng/mL) and those with 25OHD levels below the median were not statistically significant
Achiron et al., 2015 [25]	Randomized, double-blind placebo-controlled study	Multiple sclerosis patients	Alfacalcidol (1 mcg/d) for six consecutive months	80	Significant decrease in Fatigue Impact Scale (FIS) scores
**Vitamin C**					
Huck et al., 2013 [14]	Placebo-controlled pilot trial	Obese adults	Vitamin C capsule 500 mg/day or placebo for 4 weeks	20	The general fatigue score was significantly decreased in the VC group compared to the control group (*p* = 0.001)
**NADH**					
Forsyth et al., 1999 [60]	Randomized, double-blind, placebo-controlled crossover study	Chronic fatigue syndrome (CFS) patients	NADH 10 mg/day for a 4-week period	26	Statistically significant relieve of the fatigue symptoms
Santaella et al., 2004 [61]	Randomized, double-blind placebo-controlled study	Chronic fatigue syndrome (CFS) patients	NADH 5–10 mg/day for 24 months	32	Statistically significant reduction of the mean symptom score in the first trimester (*p* < 0.001). However, symptom scores in the subsequent trimesters of therapy were similar in both treatment groups.
**Thiamine**					
Sevim et al., 2017 [24]	Retrospective observational study	Multiple sclerosis patients	Sulbutiamine 400 mg/day for two months	26	Significant decrease in Fatigue Impact Scale (FIS) scores
Bager et al., 2020 [26]	Randomised, double-blinded, placebo-controlled crossover trial	Patients with inflammatory bowel disease (IBD) and severe chronic fatigue	Thiamine hydrochloride (600–1800 mg/d) for 4 weeks	40	Significant reduction in chronic fatigue

**Table 2 nutrients-15-02154-t002:** Studies employing parenteral administration.

References	Study Design	Group	Treatment	No. of Participants	Main Results
**Vitamin C**					
Suh et al., 2012 [79]	Randomized, double-blind, controlled clinical trial	Healthy volunteers	a single intravenous treatment of either vitamin C (10 g) or normal saline	141	Fatigue scores decreased significantly (*p* = 0.004) in the vitamin C group after two hours and remained lower for one day.
Takahashi et al., 2012 [76]	Multicentre prospective observational study	Cancer patients	up to 50 g IVC twice a week, for 4 weeks + oral vitamin c 2–4 g/day	60	Significant improvements in fatigue scores at 2 weeks of IVC therapy (*p* < 0.01)
Yeom et al., 2007 [77]	Prospective Observational Study	Cancer patients	10 g IVC administered twice with a 3-day interval + oral vitamin c 4 g/day for a week	39	Patients reported significantly lower scores for fatigue (*p* < 0.05)
Vollbracht et al., 2011 [78]	Epidemiological, multicentre cohort study with parallel groups	Cancer patients	i.v. vitamin C (supplied as Pascorbin^®^ 7.5 g) additional to standard tumour therapy for at least 4 weeks; control group did not receive vitamin C therapy	125	Significant reduction in fatigue symptoms
Ou et al., 2020 [75]	Single-centre, phase II, randomized clinical trial	Non-Small-Cell Lung Cancer (NSCLC) patients	1 g/kg.d IVC concurrently with modulated electrohyperthermia (mEHT) plus best supportive care (BSC), three times a week for 25 treatments; the control arm received BSC only	97	Significant reduction in fatigue symptoms evaluated by Quality-of-Life Questionnaire (QLQ-C30)
Vollbracht et al., 2018 [80]	Multicentre prospective observational study	Patients with allergy-related respiratory or cutaneous indications	i.v. vitamin C (PascorbinVR 7.5 g/50 mL) in 100 mL NaCl 0.9%	71	Significant reduction in fatigue symptoms
Jeon et al., 2016 [81]	Single-centre, randomized, double-blind, controlled clinical trial	Patients undergoing laparoscopic colectomy	IVC 50 mg/ kg bw or placebo; Single application after induction of anaesthesia	97	NS, no significant differences in fatigue score 2, 6, and 24 h post operation
**Vitamin D**					
Aziz et al., 2021 [82]	Prospective cohort study	Patients with liver cirrhosis due to chronic hepatitis C	Vitamin D3 (200,000 IU IM single dose)	50	Fatigue severity scale (FSS) and fatigue impact scale (FIS) scores improved significantly after administration of vitamin D3
**L-Carnitine**					
Brass et al., 2001 [83]	Placebo-controlled, double-blinded, randomized study	Patients with end-stage renal disease (ESRD)	Intravenous L-carnitine 10, 20, and 40 mg/kg or placebo at the conclusion of each thrice-weekly dialysis session for 24 weeks	150	Significant improvement in the fatigue domain of the Kidney Disease Questionnaire (KDQ) after 12 (*p* = 0.01) and 24 weeks (*p* = 0.03) of treatment compared with placebo
Zhang et al., 2014 [18]	Randomized, single-blind, placebo-controlled, pilot study	Metabolic syndrome (MetS) patients	4 g/day of intravenous L-carnitine for 7 days; patients in the control group were injected with saline	15	Physical fatigue (LC −3.2 ± 3.17 vs. CT 1.8 ± 2.04, *p* < 0.001) and fatigue severity (LC −11.6 ± 8.38 vs. CT 8.18 ± 7.32, *p* < 0.001) were significantly reduced in the LC group but were aggravated in the CT group
**Intravenous nutrient therapy (IVNT)**					
Massey et al., 2007 [84]	Open-label clinical trial	Therapy-resistant Fibromyalgia (FM) patients	Intravenous nutrient therapy (IVNT) once per week for 8 weeks (Modified Myers’ intravenous nutrient formula: 400 mg Magnesium chloride hexahydrate, 40 mg Calcium gluconate, 3000 mg Vitamin C, 1000 µg Hydroxocobalamin (B12), 100 mg Pyridoxine hydrochloride (B6), 250 mg Dexpanthenol (B5), 2 mg Riboflavin (B2), 100 mg Thiamine (B1), 100 mg Niacinamide	7	60% reduction in pain (*p* = 0.005) and 80% decrease in fatigue (*p* = 0.005)
**Thiamine**					
Costantini et al., 2013 [17]	Pilot study	Multiple sclerosis patients	High-dose thiamine 600–1500 mg/day orally or 100 mg/mL once a week parenterally for 20 days	15	Statistically significant relieve of the fatigue symptoms
**Methionine**					
Benedetto et al., 1993 [85]	Controlled clinical trial	Fibromyalgia (FM) patients	6 weeks of treatment with either SAMe or TENS. S-adenosyl-L-methionine (SAMe) 200-mg vial intramuscularly + two 200-mg tablets daily for 6 weeks. Patients in the TENS group (n = 15) completed five morning sessions a week	30	SAMe significantly decreased total number of tender points, pain and fatigue, and Hamilton Depression and Anxiety Rating Scales scores

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
