# Peer review of "Nutrient Therapy for the Improvement of Fatigue Symptoms"

_nutrients, 2023, doi:10.3390/nu15092154_

Round 1

Reviewer 1 Report

The article is a review of the efficacy of a nutritional intervention in patients suffering from Fatigue. The objectives are clearly stated, as well as the literature search protocol is well developed and explained in the results and discussion. The conclusions also show the most important findings of the review as well as the gaps that currently exist on this topic and the need for more extensive research in the future.

Author Response

AUTHORS: The authors would like to thank the reviewer for a thorough review of the manuscript. We would like to thank the reviewer for this valuable comment.

Reviewer 2 Report

Minor re-writing is recommended. The reasons are: There is no "Results" section. The structure of this article is 1. Introduction 2. Methods 3. Oral supplementation 4. Parenteral administration 5. Discussion 6. Conclusions I assume that 3 and 4 are the results. Some contents in 3 and 4 are recommended to move to the "Discussion" section because they have the authors' interpretation. Lines 111 to 125: recommended to move to the "Results" section.

Author Response

AUTHORS: The authors would like to thank the reviewer for this comment. A new section (Section 3. Search results) is now included in the manuscript to better highlight the results of the literature search. The collected studies are then categorised into two groups including Oral and parenteral administration routes in sections 4 and 5.

Reviewer 3 Report

The authors propose an interesting aspect of nutrition and supplementation applied to fatigue symptoms; some points should be added:

- Sarcopenia (of both the elderly and the obese) should be included in the conditions/diseases resulting in fatigue, for example, 10.3390/ijms22189724

- Creatine should certainly be included in possible supplements; there are some studies in the management of fatigue syndromes, and in every case it has a very valid rationale

- In the same way, omega3s should also be considered

- A distinction should be made between deficient elements even compared to normal conditions (such as vitamin D and zinc) and those that could have an ameliorative effect and do not have a defined RDA

- Even if, as underlined by the authors, the etiology is to be defined, in my opinion, the conditions should be distinguished if they are real pathologies or not and if they are chronic (such as fibromyalgia) or acute

- Paragraph 2.3 should be indicated as co-enzymes, even if probably for both, the prevalent action is as antioxidants

- Although difficult to unite, it would be interesting to propose a possible mechanism of action, perhaps shown in a figure

Quality of language is fine

Author Response

Referee 3

  1. Sarcopenia (of both the elderly and the obese) should be included in the conditions/diseases resulting in fatigue, for example, 10.3390/ijms22189724

AUTHORS: We would like to thank the reviewer for this valuable comment and pointing out this important condition. While, as mentioned in the suggested article, supplementation of several nutrients have been suggested to be effective in the management of sarcopenia, to our knowledge none of these studies have investigated the impact of nutrient supplementation on fatigue measures which is specifically the subject and the inclusion criteria of the current review. Namely, vitamin D has been investigated specifically for indices of sarcopenia such as muscle mass and strength. Therefore, no articles investigating fatigue symptoms in individuals suffering from sarcopenia has been identified and included in the current review. Accordingly, the following paragraph has been added in the discussion section of the manuscript.

“Moreover, several studies have investigated the role of nutrients such as vitamin D in the treatment of sarcopenia, due to the positive impact of vitamin D supplementation on muscle mass and function [97]. However, the impact of nutrient supplementation on fatigue symptoms amongst individuals suffering from sarcopenia has not been investigated. Therefore, nutrients which have been found to be associated with improvement of muscle function including vitamin D and creatine  [98], should also be specifically investigated for their potential impacts on fatigue symptoms.”

  1. Creatine should certainly be included in possible supplements; there are some studies in the management of fatigue syndromes, and in every case it has a very valid rationale.

AUTHORS: We would like to thank the reviewer for this valid point. The authors would like to highlight that nutrients which are commonly administered via both oral and parenteral routes are included in the current literature search. Therefore, while this review focuses on investigating the impact of nutrients widely available for oral and parenteral administration, future research will focus on investigating a wider range of nutritional supplements and their beneficial effects on various health outcomes.

Additionally, regarding creatine it should be noted that the current review focuses on investigating the beneficial effects of nutrient supplementation on individuals suffering from fatigue, hence studies reporting changes in fatigue scores (measured using fatigue scales such as FSS, FIS, BFI, Chalder Fatigue Scale, etc.) are included in the current review. However, to our knowledge creatine supplementation has been investigated mainly for its beneficial impacts on muscle function and investigate fatigue resistance as a measure of muscle performance and mobility, which is out of the scope of the current review [2][3]. 

Accordingly, the following lines have been included in the discussion section.

“Therefore, nutrients which have been found to be associated with improvement of muscle function including vitamin D and creatine [98], should also be specifically investigated for their potential impacts on fatigue symptoms.”

  1. In the same way, omega3s should also be considered.

AUTHORS: We would like to thank the reviewer for this valuable comment. omega-3 was not included in the literature search since the review focuses on investigating the impact of nutrients which are widely available for both oral and parenteral administration. We would like to thank the reviewer for pointing this out, future studies should explore wider range of nutritional supplements and their beneficial effects on fatigue outcomes.

  1. A distinction should be made between deficient elements even compared to normal conditions (such as vitamin D and zinc) and those that could have an ameliorative effect and do not have a defined RDA.

AUTHORS: We would like to thank the reviewer for this valid point. In order to achieve a clearer representation of the included studies the studies are now categorized into three groups for each section including studies conducted in: clinical populations, non-clinical populations and those with nutrient deficiencies. Please kindly find the main changes highlighted within the manuscript.

  1. Even if, as underlined by the authors, the etiology is to be defined, in my opinion, the conditions should be distinguished if they are real pathologies or not and if they are chronic (such as fibromyalgia) or acute.

AUTHORS: We would like to thank the reviewer for this valid point. In order to achieve a clearer representation of the included studies the studies are now categorized into three groups for each section including studies conducted in: clinical populations, non-clinical populations and those with nutrient deficiencies. Please kindly find the main changes highlighted within the manuscript.

  1. Paragraph 2.3 should be indicated as co-enzymes, even if probably for both, the prevalent action is as antioxidants.

AUTHORS: We would like to thank the reviewer for this valid comment. The section is now title as co-enzymes.

  1. Although difficult to unite, it would be interesting to propose a possible mechanism of action, perhaps shown in a figure.

AUTHORS: We would like to thank the reviewer for this comment. While the mechanism of action of micronutrients have been briefly mentioned within the manuscript, unfortunately having a figure representing the mechanisms of action of all the nutrients would not be possible. 

References

[1]        R. Cannataro et al., ‘Sarcopenia: Etiology, Nutritional Approaches, and miRNAs’, IJMS, vol. 22, no. 18, p. 9724, Sep. 2021, doi: 10.3390/ijms22189724.

[2]        Dolan, Artioli, Pereira, and Gualano, ‘Muscular Atrophy and Sarcopenia in the Elderly: Is There a Role for Creatine Supplementation?’, Biomolecules, vol. 9, no. 11, p. 642, Oct. 2019, doi: 10.3390/biom9110642.

[3]        S. A. Smith, S. J. Montain, R. P. Matott, G. P. Zientara, F. A. Jolesz, and R. A. Fielding, ‘Creatine supplementation and age influence muscle metabolism during exercise’, Journal of Applied Physiology, vol. 85, no. 4, pp. 1349–1356, Oct. 1998, doi: 10.1152/jappl.1998.85.4.1349.

Round 2

Reviewer 3 Report

The authors made a nice improvement of the manuscript so is ready to be published

it is fine